# In Situ 3D Scene Synthesis for Ubiquitous Embodied Interfaces

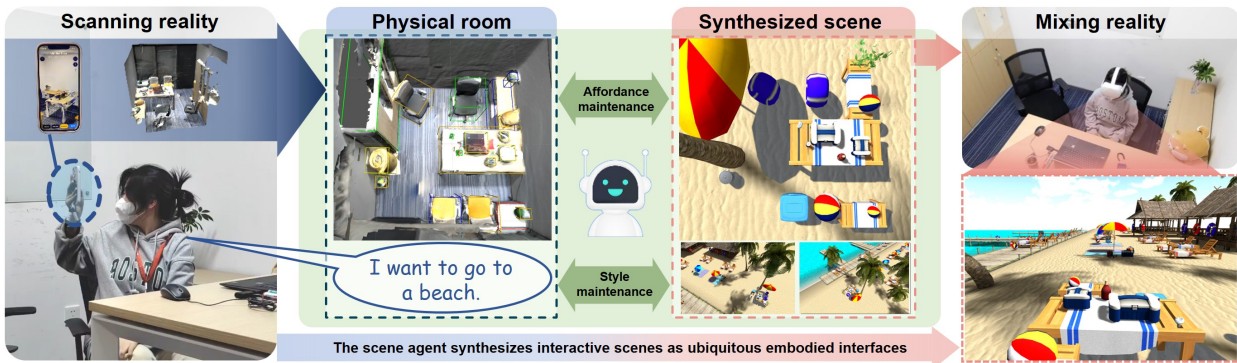

**Figure 1: We propose a scene agent to synthesize virtual scenes by observing the situated physical environment and the user's demand represented by language. The synthesized scenes maintain the affordance of physical objects and maintain the style described by the user, enhancing users' sense of security and interactive experience in VR. This technique contributes to building ubiquitous embodied interfaces for users to conveniently enter the virtual world.**

## ABSTRACT

Virtual reality (VR) provides an interface to access virtual environments anytime and anywhere, allowing us to experience and interact with an immersive virtual world. It has been widely used in various fields, such as entertainment, training, and education. However, the user's body cannot be separated from the physical world. When users are immersed in virtual scenes, they encounter safety and immersion issues caused by physical objects in the surrounding environment. Although virtual scene synthesis has attracted widespread attention, many popular methods are limited to generating purely virtual scenes independent of the physical environment or simply mapping all physical objects as obstacles. To this end, we propose a scene agent that synthesizes situated 3D virtual scenes as a kind of ubiquitous embodied interface in VR for users. The scene agent synthesizes scenes by perceiving the user's physical environment as well as inferring the user's demands. The synthesized scenes maintain the affordances of the physical environment, enabling immersive users to interact with the physical environment and improving the user's sense of security. Meanwhile, the synthesized scenes maintain the style described by the user, improving the user's immersion. The comparison results show that the proposed scene agent can synthesize virtual scenes with better affordance maintenance, scene diversity, style maintenance, and 3D intersection over union (3D IoU) compared to state-of-the-art baseline methods. To the best of our knowledge, this is the first work

that achieves in situ scene synthesis with virtual-real affordance consistency and user demand.

## CCS CONCEPTS

• **Human-centered computing → Mixed / augmented reality**; **Virtual reality**.

## KEYWORDS

Scene synthesis, affordance, user demand, large language model.

## 1 INTRODUCTION

VR has the potential to enhance the physical environment, extending the boundaries of the physical world [22] and providing a highly interactive and immersive environment for users in a variety of applications (*e.g.*, games, training, and education). This enables users to experience a variety of environments from a single physical location, thereby alleviating the need to travel, reducing carbon emissions, enhancing productivity, and potentially augmenting overall life satisfaction [32, 53]. For example, many works have provided virtual offices for knowledge workers [3, 23] to improve their working experience. Most of the current virtual scenes are set manually by professionals. Fortunately, recent progress in scene synthesis makes it possible to acquire low-cost and high-quality virtual scenes. The 3D models-based method is an efficient way to synthesize scenes [35, 61] which has a wide range of applications, from indoor design and games to simulators for embodied artificial intelligence (AI). However, since human users are always located in physical environments, an important problem arises in implementing virtual applications: How to acquire virtual scenes that are consistent with the constraints of physical space?

On the one hand, most VR applications are used in indoor rooms without enough space. A common risk is that users might hit objects around them when using VR devices [12, 30]. On the other hand, physical objects can offer VR users affordances [31, 40, 55]

*ACM MM, 2024, Melbourne, Australia*
© 2024 Copyright held by the owner/author(s). Publication rights licensed to ACM.
ACM ISBN 978-x-xxxxx-xxxx-x/YY/MM
https://doi.org/10.1145/nnnnnnn.nnnnnnn

and passive feedback [19, 20, 28, 57] that enhance their experience and even enhance the interactivity of the virtual environment. Leveraging the affordances of physical objects could improve user experience and task performance in VR applications [10, 11, 26, 28]. Therefore, some works synthesize virtual scenes based on physical environments [9, 56, 59, 65]. However, these works usually adopt 3D models that are consistent with physical objects [56] (*e.g.*, virtual tables for physical tables), reducing the diversity of virtual scenes. Alternatively, these works may only consider walking areas [9, 59, 65] while neglecting other interactions between users and environments.

In the context of situated scene synthesis, the main goal is to synthesize interactive scenes based on their situated physical environments, considering the affordance of the physical objects. When the user is immersed in the virtual scene, they can also perceive and utilize the affordance of physical objects, which guarantees a highly immersive experience for users. At the same time, it is also necessary to synthesize the virtual scenes that can satisfy users' demands. Generally, users can be immersed in any virtual scene they want if the synthesized scenes are unlimited, so understanding the user's demand is important for meeting personalized requirements. Due to the uncertainty of physical environments and the various personalized needs of users, a well-situated scene synthesis solution should not only exploit the physical objects as building blocks for better physical-virtual consistency but also understand human users' demands via efficient interactions, such as natural language [39, 58].

Therefore, we propose a scene agent, which leverages the large language model's ability of information extraction [33] and its prior knowledge related to scenes [16], that can observe both user demands and the situated physical environments to synthesize virtual scenes with interactivity, as shown in figure 1. For each physical object, the scene agent infers the corresponding virtual object by considering two aspects. One is the affordance similarity between physical objects and virtual objects. Another is the style similarity (including place, season, and object) between user demands and virtual objects. Afterward, according to the physical information, the scene agent synthesizes scenes by translating, rotating, and scaling virtual objects. The synthesized scene can not only maintain the same affordances of physical objects but also maintain the scene style that the user wants.

To the best of our knowledge, this is the first work that synthesizes arbitrary virtual scenes with physical interactivity considering both the physical environment and the user's demand. Overall, our contributions are threefold:

(1) We propose a language model-based 3D scene synthesis method to extract information of the physical environment, virtual objects, and the user input text, generating their semantic relations for building the interactive agent system.

(2) We develop a scene agent based on the above method to perceive the physical affordance and user demand, which can synthesize interactive virtual scenes for handling physical constraints and satisfying the user's personalized demands.

(3) We conduct comparison studies between our method and three baselines, followed by a perceptual study, to demonstrate that the proposed scene agent could synthesize better scenes with affordance maintenance and style maintenance.

## 2 RELATED WORK

### 2.1 3D Scene synthesis

Generative models have contributed to the synthesis of outdoor 3D scenes [63]. However, these generated scenes do not support human-object interaction. Researchers synthesize indoor 3D scenes by selecting objects from object datasets and generating layouts based on procedural modeling with grammars [34, 47, 50], graph [35, 38, 61, 68], auto-regressive neural networks [52], transformer [46], and diffusion models [15]. Some methods consider the interaction between humans and environments, such as human motions[51] with human-object contact [66] and poses [67]. Affordance could serve as a bridge to characterize the human-object relations[51]. The aforementioned human-centric scene synthesis methods all took advantage of the object affordance for human-object interaction. However, these works are based on existing human interaction actions. In this work, we will synthesize virtual scenes considering object affordances without human action priors.

### 2.2 Language-driven 3D Scene synthesis

Language, as an important medium for human-computer interaction, has been used for 3D scene synthesis.RoomDreamer [58] aligned the geometry and texture to the input scene structure and prompt simultaneously. GAUDI [2] was a generative model that enabled both unconditional and conditional synthesis including image, text, category, etc. SceneDreamer [8] synthesized unbounded in-the-wild 3D scenes from 2D images based on the GAN network. CTRL-ROOM [15] controlled the scene synthesis with a diffusion model, enabling the change of scenes. However, those synthesized scenes cannot support immersive interaction in VR. PiGraphs [54] synthesized human pose priors-based scenes that only included human action-related objects mentioned in the language specifications. Chang *et al.* [4–7] and Ma *et al.* [39] parsed the input text into a knowledge tree or graph for synthesis, where the initial scene could be changed by language. These methods aim at indoor scene synthesis, and the input texts need explicit instructions and can only directly specify virtual objects in the database. In this work, we plan to synthesize both indoor and outdoor virtual scenes without explicit instructions.

### 2.3 Situation-aware scene synthesis

Some works synthesize scenes based on the user-situated physical world. Human-in-loop paradigm-based methods require users to manually place virtual objects in the positions of corresponding physical objects [14, 29]. Other methods adopt the auto-generation paradigm. DreamWalker [65] detected walkable paths and obstacles. The paths are mapped as resembling paths in the virtual scene and the obstacles as default virtual objects. VRoamer [9] and Sra *et al.* [59, 60] extracted walkable areas and physical obstacles according to the scanned physical environment. VRoamer generated corridors and doors for walkable areas while bricks or spikes for obstacles. Sra *et al.* [59, 60] generated boundary elements in the boundary of the walkable areas, where several special objects (*e.g.*chairs) were mapped as virtual counterparts to leverage the affordance of the physical objects.

Those methods only consider the walkable areas and several special objects, leading to a lack of interactivity in the synthesized 3D scene. Shapira [55] proposed a method that first placed specialized 3D object models in the scene and then optimized their arrangement based on planar areas without considering the interactivity. Our goal is to generate arbitrary scenes which have the same interactivity as the physical world.

## 3 PRELIMINARY

In this section, we introduce the concepts and symbols adopted.
**Scene presentation.** A physical environment $\mathcal{S}_{phy}$ including all $N$ physical object information $\left\{o_n^{phy}\right\}_{n=1}^N \in O^{phy}$, where each tuple $o_n^{phy}$ denotes the information of a physical object. A synthesized virtual scene $\mathcal{S}_{vir}$ including a basic scene background $b_i^{vir}$ and all $M$ virtual object information $\left\{o_m^{vir}\right\}_{m=1}^M \in O^{vir}$, where each tuple $o_m^{vir}$ denotes the information of a virtual object. $\mathcal{B} = \left\{b_h^{vir}\right\}_{h=1}^H$ denotes all $H$ basic scenes. For a physical object $o_i^{phy}$ or a virtual object $o_i^{vir}$, it contains attribute information $\{c_i, d_i, t_i, r_i, s_i\}$: category $c_i$, description $d_i$ (can be empty), bounding box location $t_i = (tx_i, ty_i, tz_i) \in \mathbb{R}^3$, bounding box rotation $r_i = (rw_i, rx_i, ry_i, rz_i) \in \mathbb{R}^4$, and bounding box size $s_i = (sx_i, sy_i, sz_i) \in \mathbb{R}^3$.

**Affordance.** Affordance was first introduced by psychologist Gibson [18], which represents the action possibilities of an object that are perceivable by an actor [25]. $\mathcal{A} = \{a_k\}_{k=1}^K$ is a tuple of $K$ kind of affordances of a object, where each tuple $a_k$ denotes one kind of affordance. Based on previous works [25, 37], we consider ten different affordances for the objects, including *walkable, supportable, sitable, drinkable, eatable, graspable, breakable, dangerous, moveable, obstructive*.

**Virtual place type** Theoretically, the types of virtual places can be unlimited. We ask GPT-4 [45] to summarize P virtual place types that people want to go to. We find that when P>20, the types are repeated. Therefore, we select 20 types: *Library, Conservatory, Spa, Lounge, Observatory, Suite, Monastery, Studio, Bookstore, Aquarium, Beach, Forest, Garden, Vineyard, Yacht, Rooftop, Treehouse, Reef, Peak, Rainforest*. $\mathcal{E} = \left\{e_p\right\}_{p=1}^P$ denotes the tuple of $P$ kinds of virtual places, where $e_p$ denotes the $p$-th virtual place.

**Season** Some objects have obvious seasonality, such as a bench covered by snow. We consider the probability that each object can appear in each reason including *spring, summer, autumn, winter*. $\mathcal{T} = \left\{t_j\right\}_{j=1}^J$ denotes the tuple of $J$ kinds of seasons, where $t_j$ denotes the $j$-th season.

**User demand** Users could express what kind of scene they want to go by a sentence $u$. Speech is also compatible as it can be converted to texts by speech-to-text techniques. In this paper, we extract **season** $t^{user}$, **place** type $e^{user}$, and possible **objects** information from user demand. $O^{user} = \left\{o_q^{user}\right\}_{q=1}^Q$ denotes a tuple of $Q$ kinds of objects mentioned. $o_q^{user}$ only includes the category $c$.

## 4 SYNTHESIS METHOD

Our proposed scene agent synthesizes scenes by observing the situated physical environment and the user's demand. The physical environment information could be obtained via volumetric instance-aware semantics mapping methods from RGB-D information[21, 24]. Our goal is to synthesize user-expected virtual scenes. These scenes maintain the affordances of the physical environment and maintain the style that the user demands. Formally, guided by a physical environment $\mathcal{S}_{phy}$ and a user demand $u$, the proposed scene agent synthesizes scenes $\mathcal{S}_{vir} \sim \mathcal{P}(\mathcal{S}_{vir}|\mathcal{S}_{phy}, u)$.

Synthesizing a virtual scene with one sentence is a hefty task. Therefore, apart from plain text from the user, we also consider information about the physical environment and virtual objects to solve this problem. The first step is to understand the user's demand. At the same time, we infer the affordance of the situated physical environment. In addition, we infer the features of all virtual objects. Finally, we synthesize the whole virtual scene based on all the results of the first three steps. The whole scene synthesis pipeline is shown in figure 2 and the algorithm is outlined in Algorithm 1.

---

**Algorithm 1** In situ scene synthesis

---

**Input:** $u$: user's input; $\mathcal{A}$: all affordances; $\mathcal{E}$: all virtual places; $\mathcal{T}$: all seasons; $O^{vir}$: all virtual objects, $o_m^{vir} \in O^{vir}$; $O^{phy}$: all physical objects, $o_n^{phy} \in O^{phy}$; $E_{user}(e^{user}, t^{user}, O^{user}|u)$: text extractor; $L(\cdot, \cdot)$: language similarity; $A(V_{\mathcal{A}}|\mathcal{A}, O)$: affordance predictor; $E(V_{\mathcal{E}}|O)$: place predictor; $T(V_{\mathcal{T}}|O)$: season predictor.

**Output:** $\mathcal{S}_{vir}$: the synthesized virtual scene corresponding with the physical environment $\mathcal{S}_{phy}$.

//* User demand inference.*//
1: $\{e^{user}, t^{user}, O^{user}\} \sim E_{user}(\cdot|u)$
2: $V_{\mathcal{B}^{user}} \leftarrow L(e^{user}, \mathcal{B}), V_{\mathcal{B}^{user}} \in \mathbb{R}^H$
3: $V_{\mathcal{E}^{user}} \leftarrow L(e^{user}, \mathcal{E}), V_{\mathcal{E}^{user}} \in \mathbb{R}^P$
4: $V_{\mathcal{T}^{user}} \leftarrow L(t^{user}, \mathcal{T}), V_{\mathcal{T}^{user}} \in \mathbb{R}^J$
5: **for** $q = 1 : Q$ **do**
6: $\quad V_{o_q^{user}} \leftarrow L(o_q^{user}, O^{vir}), o_q^{user} \in O^{user}, V_{o_q^{user}} \in \mathbb{R}^M$
7: **end for**
8: $V_{O_Q^{user}} = [V_{o_1^{user}}, V_{o_1^{user}}, ..., V_{o_Q^{user}}], V_{O_Q^{user}} \in \mathbb{R}^{Q \times M}$
9: $V_{O^{user}} = \max_i V_{O_Q^{user}}[i, j], V_{O^{user}} \in \mathbb{R}^M$
//* Physical affordance inference.*//
10: **for** $n = 1 : N$ **do**
11: $\quad V_{\mathcal{A}_n^{phy}} \sim A(\cdot|\mathcal{A}, c_n, s_n), \{c_n, s_n\} \in o_n^{phy}, V_{\mathcal{A}_n^{phy}} \in \mathbb{R}^K$
12: **end for**
//* Virtual object-based inference.*//
13: **for** $m = 1 : M$ **do**
14: $\quad V_{\mathcal{A}_m^{vir}} \sim A(\cdot|\mathcal{A}, c_m, s_m), \{c_m, s_m\} \in o_m^{vir}, V_{\mathcal{A}_m^{vir}} \in \mathbb{R}^K$
15: $\quad V_{\mathcal{E}_m^{vir}} \sim E(\cdot|\mathcal{E}, c_m, d_m), \{c_m, d_m\} \in o_m^{vir}, V_{\mathcal{E}_m^{vir}} \in \mathbb{R}^P$
16: $\quad V_{\mathcal{T}_m^{vir}} \sim T(\cdot|\mathcal{T}, c_m, d_m), \{c_m, d_m\} \in o_m^{vir}, V_{\mathcal{T}_m^{vir}} \in \mathbb{R}^J$
17: **end for**
//* Whole scene synthesis.*//
18: $b_i^{vir} \leftarrow \arg\max V_{\mathcal{B}^{user}}$
19: **for** $n = 1 : N$ **do**
20: $\quad o_n^{vir} \leftarrow \arg\max_{o_m^{vir} \in O^{vir}} (C(V_{\mathcal{A}_n^{phy}}, V_{\mathcal{A}_m^{vir}}) * C(V_{\mathcal{E}^{user}}, V_{\mathcal{E}_m^{vir}}) * C(V_{\mathcal{T}^{user}}, V_{\mathcal{T}_m^{vir}}) * S(o_n^{phy}, o_m^{vir}) * (V_{O^{user}}[m]))$
21: $\quad \{t_n, r_n, s_n\}$ of $o_n^{vir} \leftarrow IoU(o_n^{vir}|o_n^{phy})$
22: **end for**
23: $\mathcal{S}_{vir} \leftarrow$ Load $b_i^{vir}$ and all $\left\{o_n^{vir}\right\}_{n=1}^N$

---

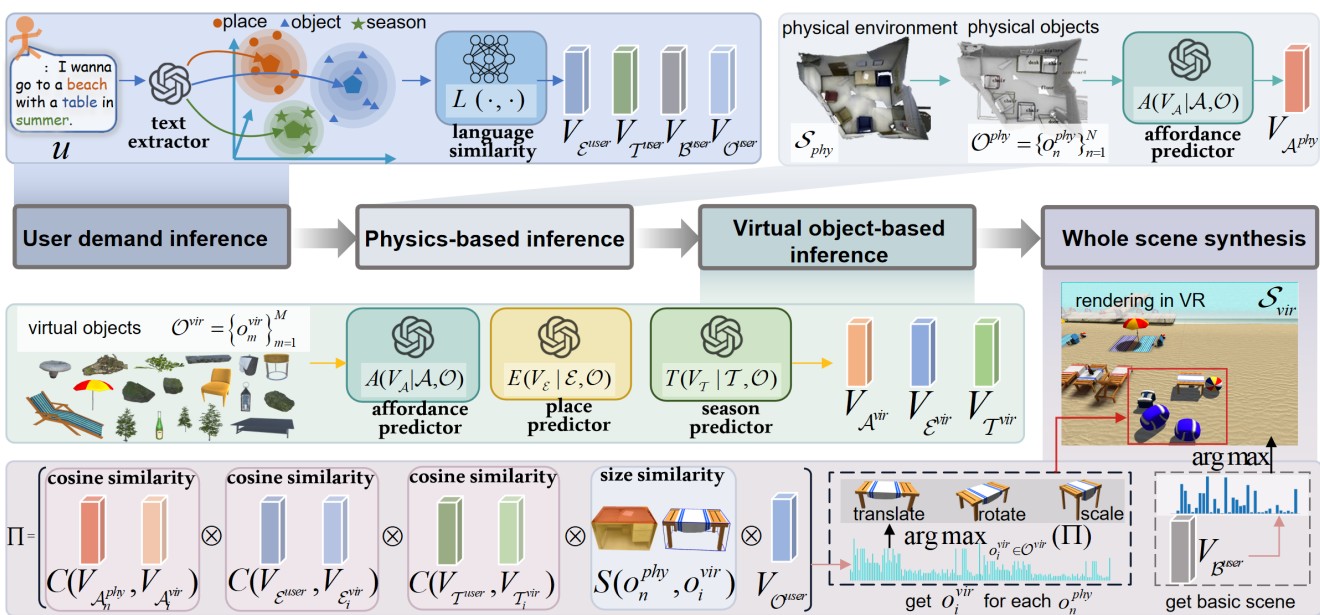

**Figure 2: The synthesis algorithm.** First, the LLM extracts the place, season, and objects mentioned by the user, which are used to predict the similarity with all places, seasons, and virtual objects via a language similarity module. Moreover, the affordance confidence of each physical object as well as the attribute likelihoods of virtual objects are predicted by the LLM. Next, a cosine similarity module is used to calculate the three similarities: the affordance similarity between each physical object and all virtual objects; the place and season similarities between the virtual objects and the user demand. A size similarity module calculates the size similarity between each physical object and all virtual objects. Finally, the corresponding virtual objects with the highest likelihood for each physical object and the basic scene with the highest likelihood are used for scene synthesis. ⊗ means the element-wise multiplication of the vectors of the likelihoods.

## 4.1 User demand inference

We propose a user text extractor $E_{user}(e^{user}, t^{user}, O^{user}|u)$ based on a large language model (LLM) model which infer the **place** $e^{user}$, **season** $t^{user}$, and possible **objects** information $O^{user}$ according to the prompt with the user demand $u$. $e^{user}$, $t^{user}$ and $O^{user}$ can be empty. More details about $E_{user}(e^{user}, t^{user}, O^{user}|u)$ can be found in the Appendix.

We adopt the similarity predictor $L(\cdot, \cdot)$ [17] to infer: the scene background similarity $V_{\mathcal{B}^{user}} \in \mathbb{R}^H$ between user mentioned place $e^{user}$ and each basic scene $b_i^{vir}$ in $\mathcal{B}$ by $L(e^{user}, \mathcal{B})$; the similarity $V_{\mathcal{E}^{user}} \in \mathbb{R}^P$ between user mentioned place $e^{user}$ and each place $e_k$ in $\mathcal{E}$ by $L(e^{user}, \mathcal{E})$; the similarity $V_{\mathcal{T}^{user}} \in \mathbb{R}^J$ between user mentioned season $t^{user}$ and each season $t_k$ in $\mathcal{E}$ by $L(t^{user}, \mathcal{T})$; the similarity $V_{O^{user}} \in \mathbb{R}^M$ between all virtual objects $O^{vir}$ and user mentioned objects $O^{user}$. $V_{O^{user}}$ represents the maximum value of similarity between each virtual object and all objects mentioned by the user. $V_{O^{user}} = \max_i V_{O_Q^{user}}[i, j]$ and $V_{O_Q^{user}} = [V_{o_1^{user}}, V_{o_2^{user}}, ..., V_{o_q^{user}}]$, $V_{O_Q^{user}} \in \mathbb{R}^{Q \times M}$ and $V_{O_Q^{user}}$ is the similarity matrix of all virtual objects and all objects mentioned in user text $u$. Specially, if $e^{user}$, $t^{user}$ or $O^{user}$ is empty, all values of corresponding similarity in $V_{\mathcal{B}^{user}}$, $V_{\mathcal{E}_{user}}$, $V_{\mathcal{T}_{user}}$ or $V_{O^{user}}$ are 1.

## 4.2 Physics-based inference

For synthesizing scenes where users could take advantage of physical objects' affordance, the affordance of the virtual object should be aligned with that of the corresponding physical object. Therefore, we propose an affordance LLM-based predictor $A(V_{\mathcal{A}}|\mathcal{A}, O)$ to infer the confidence of each affordance of an object. For the $n$-th physical object, the affordance confidence $V_{\mathcal{A}_n^{phy}}$ can be got by $A(\cdot|\mathcal{A}, c_n, s_n), \{c_n, s_n\} \in o_n^{phy}$ according to the prompt with the category $c_n$ and size $s_n$ of the $n$-th physical object and the affordance list $\mathcal{A}$. All confidences are represented by a one-dimensional vector. More details about $A(V_{\mathcal{A}}|\mathcal{A}, O)$ can be found in the Appendix.

## 4.3 Virtual object-based inference

We use the affordance predictor $A(V_{\mathcal{A}}|\mathcal{A}, O)$ to infer the confidence of each affordance of a virtual object. For the $m$-th virtual object, the affordance confidence $V_{\mathcal{A}_m^{vir}}$ can be got by $A(\cdot|\mathcal{A}, c_m, s_m), \{c_m, s_m\} \in o_m^{vir}$ according to the prompt with its category $c_m$, its size $s_m$ and the affordance list $\mathcal{A}$. Additionally, we propose a place predictor $E(V_{\mathcal{E}}|\mathcal{E}, O)$ based on the LLM to infer the likelihood of a virtual object appearing in each virtual place. For the $m$-th virtual object, its likelihood of appearing in each place $E(V_{\mathcal{E}}|\mathcal{E}, O)$ can be got by $E(\cdot|\mathcal{E}, c_m, s_m, d_m), \{c_m, d_m\} \in o_m^{vir}$ according to the prompt with its category $c_m$ and description $d_m$. Moreover, we propose a season predictor $T(V_{\mathcal{T}}|\mathcal{T}, O)$ based on the LLM to infer the likelihood of a virtual object appearing in each season. For $m$-th virtual object, the likelihood of it appearing in each season $V_{\mathcal{T}_m^{vir}}$ can be got by $T(\cdot|\mathcal{T}, c_m, s_m), \{c_m, s_m\} \in o_m^{vir}$ according to the prompt with its

category $c_m$ and description $d_m$. Similarities, confidences, and likelihoods are represented by one-dimensional vectors. More details can be found in the Appendix.

## 4.4 Scene synthesis

As shown in Algorithm 1, after getting the results of Section 4.1, 4.2 and 4.3, the whole virtual scene can be synthesized. First, we get the basic scene $b_i^{vir}$ corresponding to the maximum value in the basic scene likelihood $V_{\mathcal{B}_{user}}$. We then get the corresponding virtual object $o_n^{vir}$ for each physical object through the maximum value considering two aspects. One is the affordance and size similarities between virtual objects and physical objects. Another is the place, season, and object similarities between virtual objects and the user demand. We propose a module $C(V_i, V_j)$ to calculate the cosine similarity of two vectors and a module $S(o_i, o_j)$ to calculate the size similarity of virtual objects and physical objects. More details about the $S(o_i, o_j)$ can be found in the Appendix. Additionally, we propose an adjusted module $IoU(o^{vir}|o^{phy})$ to get the position, rotation, and size $\{t_n, r_n, s_n\}$ of each virtual object corresponding to the physical object. $IoU(o^{vir}|o^{phy})$ minimizes the value of 3D Intersection over Union (IoU) between the virtual object $o_n^{vir}$ and the physical object $o_n^{phy}$ by adjusting the position, rotation, and size of the virtual object. Finally, we load the selected basic scene $b_i^{vir}$ and all selected virtual objects $\{o_n^{vir}\}_{n=1}^N$ to get the synthesized scene $S_{vir}$. More details can be found in the Appendix.

## 5 EXPERIMENT SETUP

### 5.1 Dataset and Implementation Details

To evaluate the synthesis performance of the proposed method, we use 12 indoor scenes from [13] and 18 scenes from [27] as the physical room input. For each physical room, 12 sentences of user demand are used for scene synthesis. In total, $(12 + 18) * 12 = 360$ synthesized scenes are used for evaluation in the experiments. In addition, a total of 350 virtual objects from three packages in the unity asset store [1, 42, 43] are organized for scene synthesis. Figure 3 demonstrates several synthesized scenes.

### 5.2 Baselines

We compare the proposed method with three typical baselines: **LLM**, **Semantics**, and **VRoamer**-based methods. Similar to Feng's work [16], the LLM-based method predicts the corresponding virtual object for each physical object using its information as the prompt. Similar to the methods built-in commercial head-mounted displays [41, 49] which deploy the virtual objects with the same category of physical objects, the Semantics-based method predicts the corresponding virtual object for each physical object based on the language similarity between the virtual and physical objects. VRoamer-based method [9] synthesizes the scene by using virtual objects with *obstructive* affordance. In our VRoamer-based baseline, the virtual objects with a 1.0 confidence of *obstructive* affordance are randomly used to synthesize the scene. In addition, LLM-based, Semantics-based, and VRoamer-based without (w/o) size (**LLM w/o size, Semantics w/o size, VRoamer w/o size**) or with size (**LLM with size, Semantics with size, VRoamer with size**) constraints

respectively are compared. Figure 4 demonstrates examples synthesized by different methods. Please find more details about the baselines in the Appendix.

**Table 1: Results of quantitative comparison on SceneNN dataset.**

| Methods | KL Div.(↓) | SD (↑) | Sty. Sim.(↑) | 3D IoU (↑) w/ scale | w/o scale |
|---|---|---|---|---|---|
| LLM w/o size | 0.198 | 0.134 | 0.550 | 0.362 | 0.114 |
| LLM with size | 0.748 | 0.218 | 0.554 | 0.705 | 0.348 |
| Semantics w/o size | 0.149 | 0.319 | 0.500 | 0.486 | 0.173 |
| Semantics with size | 0.217 | 0.361 | 0.512 | 0.743 | 0.356 |
| VRoamer w/o size | 0.455 | 0.134 | 0.571 | 0.349 | 0.109 |
| VRoamer with size | 0.327 | 0.184 | 0.568 | 0.660 | 0.322 |
| Ours | **0.027** | **0.386** | **0.763** | **0.858** | **0.427** |

**Table 2: Results of quantitative comparison on ProcTHOR dataset.**

| Methods | KL Div. (↓) | SD (↑) | Sty. Sim.(↑) | 3D IoU (↑) w/ scale | w/o scale |
|---|---|---|---|---|---|
| LLM w/o size | 1.057 | 0.168 | 0.533 | 0.208 | 0.057 |
| LLM with size | 0.364 | 0.311 | 0.542 | 0.620 | 0.302 |
| Semantics w/o size | 0.206 | 0.493 | 0.513 | 0.409 | 0.148 |
| Semantics with size | 0.033 | 0.545 | 0.521 | 0.684 | 0.338 |
| VRoamer w/o size | 1.196 | 0.207 | 0.578 | 0.183 | 0.051 |
| VRoamer with size | 0.854 | 0.246 | 0.570 | 0.381 | 0.180 |
| Ours | **0.042** | **0.618** | **0.749** | **0.729** | **0.368** |

### 5.3 Quantitative evaluation

For indoor scene synthesis, Kullback-Leibler (KL) Divergence between the category distribution of predicted and ground truth scenes and the FID scores of specific projection [16, 48, 52] are usually adopted as evaluation metrics. Our approach synthesizes different scenes without ground truth rather than indoor scenes, leading to the above two metrics are not suitable for evaluating our approach. One of our goals is to synthesize scenes with the same affordance as physical environments. Furthermore, our approach is expected to synthesize scenes containing different types of virtual objects as in physical environments. Therefore, we measure the **affordance maintenance** and **scene diversity** of the synthesized scenes. In addition, we measure **style similarity** between the virtual objects in the synthesized scenes and the user demand and the **3D intersection over union (IoU)** between the physical objects and the corresponding virtual objects. Table 1 and Table 2 show the comparison results compared with six baselines on SceneNN dataset [27] and ProcTHOR dataset [13].

*5.3.1 Affordance maintenance.* We measure the affordance maintenance via KL Divergence between the affordance class distribution of the virtual objects and the affordance class distribution of the physical objects. The results show that the objects in the scene synthesized by our method have more consistent affordances with physical objects than the baselines.

*5.3.2 Scene diversity.* We measure the scene diversity (SD) via the number of object type distances between the types of objects in synthesized scenes and in physical environments.

$$SD = 1 - \frac{|N_{syn} - N_{phy}|}{N_{phy}} \quad (1)$$

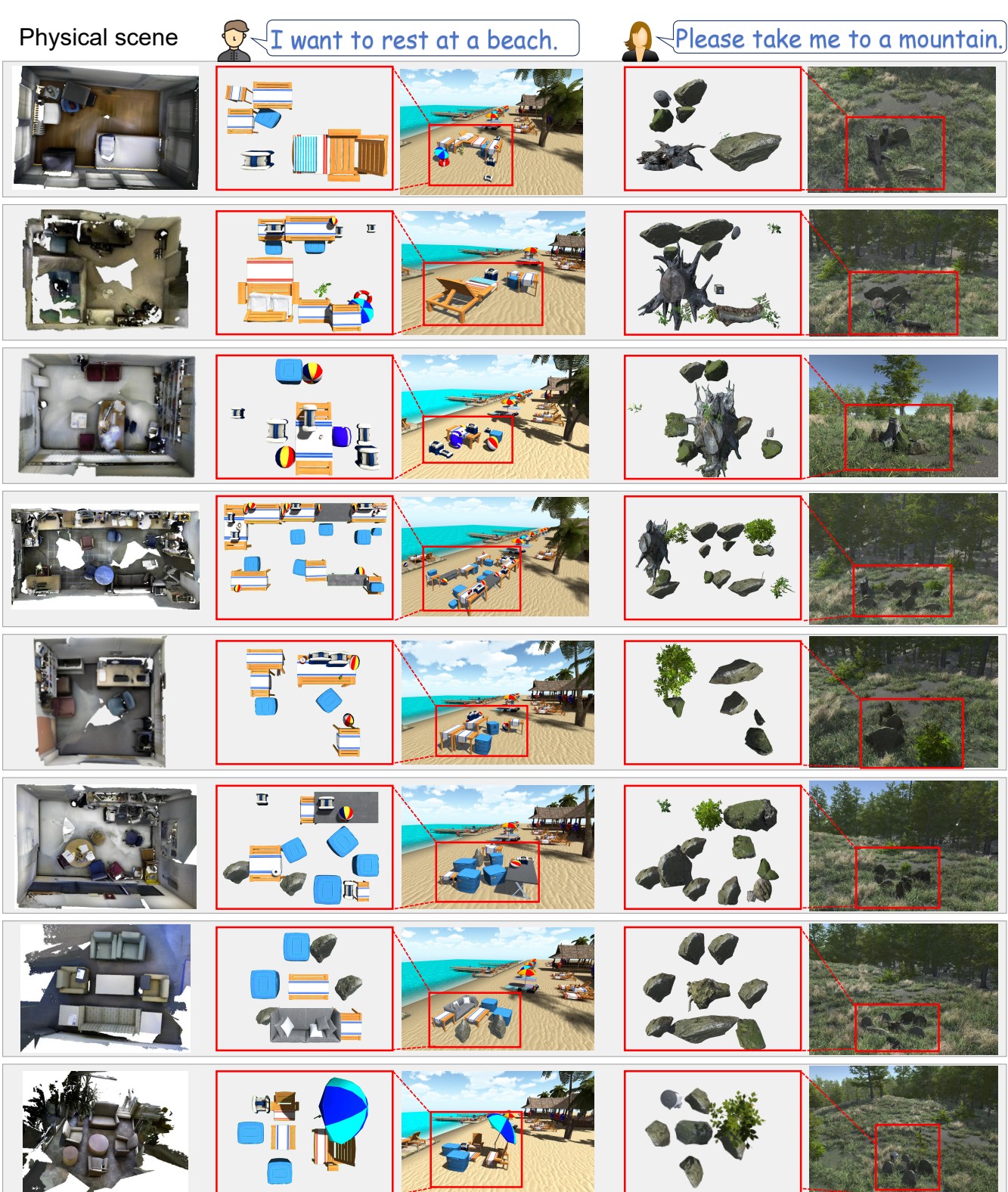

Figure 3: The examples of the synthesized scenes of our method.

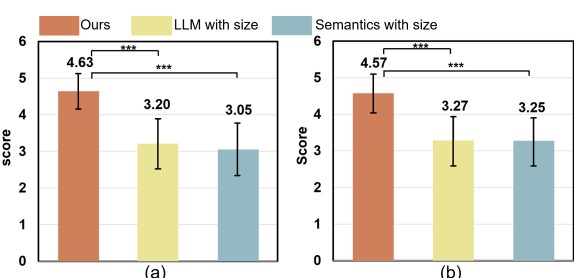

**Figure 4: The examples of the synthesized scenes of four methods.**

where $N_{syn}$ means the number of types of objects in the synthesized scenes and $N_{phy}$ means that in the physical environments. The bigger value of SD means that synthesized scenes and physical environments have a similar number of object types. The results show that the scenes synthesized by our method have a more similar number of object types to the physical environments compared with the baselines. That means scenes synthesized by our method have a more realistic scene diversity.

*5.3.3 Style similarity.* We hope the style of virtual objects can meet the user's demands. Therefore, we measure the style similarity (Sty. Sim.) between virtual objects in synthesized scenes and user input.

$$Sty.\ Sim. = w_1 * Average(\underbrace{\sum_{n=1}^{N}(V_{B^{user}}[ind\_m] * V_{E_n^{vir}}[ind\_m])}_{scene\ similarity})$$

$$+ w_2 * Average(\underbrace{\sum_{n=1}^{N} C(V_{\mathcal{T}^{user}}, V_{\mathcal{T}_{ind\_n}^{vir}})}_{season\ similarity}) + w_3 * Average(\underbrace{\sum_{n=1}^{N} V_{O_{ind\_n}^{user}}}_{object\ similarity})$$

(2)

where $w_1 = \frac{1}{3}$, $w_2 = \frac{1}{3}$, and $w_3 = \frac{1}{3}$ are the weights of the scene similarity, season similarity, and object similarity. $ind\_m$ is the index satisfying $V_{B^{user}}[ind\_m] = \arg\max(V_{B^{user}})$. $V_{B^{user}}[ind\_m]$ means the $ind\_m$-th scene background that best matches user demand. $V_{E_n^{vir}}[ind\_m]$ means the likelihood of the $n$-th object appearing in the $ind\_m$-th scene. $ind\_n$ is the index of virtual objects corresponding to the $n$-th physical objects. $C(V_{\mathcal{T}^{user}}, V_{\mathcal{T}_{ind\_n}^{vir}})$ means the similarity between the season likelihood of $ind\_n$-th virtual object matching the seasons mentioned by the users in their demand $u$. $V_{O_{ind\_n}^{user}}$ means the likelihood of the $ind\_n$-th virtual objects matching the objects mentioned in the user demand $u$. The results show that the scenes synthesized by our method can better maintain the style of the scene that the user demands compared to the baselines.

*5.3.4 3D IoU.* We measure the 3D intersection over union (IoU) to evaluate the degree of overlap between virtual objects of synthesized scenes and physical objects in physical environments. We compare all methods in two situations: *with (w/) scale* and *without (w/o) scale*. *with (w/) scale* means the virtual objects are scaled according to the size of the physical objects, while *without (w/o) scale* means the virtual objects keep the size of themselves. The scaling

factor is limited to the range from 0.5 to 2 to avoid deforming objects too much. The results show that the scenes synthesized by our method have better 3D IoU.

## 5.4 Qualitative experiment

**Figure 5: The result of the perceptual study. (a) The scores that the synthesized scene matches the user's description; (b) Affordance and style maintenance of the synthesized scenes. (*** means $p < 0.001$.)**

We conduct a perceptual study to evaluate the quality of the synthesized scenes as [46]. To this end, we randomly sampled 6 scenes for evaluation. 10 subjects participated in the study. The results of LLM-based without size and semantics-based without size have huge errors. VRoamer-based methods adopt random obstacles to synthesize scenes, which has poor results. Therefore, we compared our methods with **LLM with size** and **Semantics with size** in this perceptual study. Participants filled out scores of the following two questions on a 5-point Likert scale (1 is the least consistent and 5 is the most consistent) for each scene. A total of 60 sets of data are collected. **Q1:** The synthesized scene matches the user's demands. **Q2:** The objects in the synthesized scene maintain affordances to the objects in the physical room and maintain style consistency.

Figure 5 shows the results of the two questions. General repeated measures ANOVA tests and paired T-tests with correction, if needed, are used to analyze the data. There is a significant difference among the three groups (Q1: $F_{2,46.439} = 101.876$, $p < 0.001$; Q2: $F_{2,39.627} = 100.338$, $p < 0.001$). The scenes synthesized by our method are significantly better than **LLM with size** (Q1: $t_{59} = 12.672$, $p < 0.001$, Q2: $t_{59} = 13.105$, $p < 0.001$ ) and **Semantics with size** (Q1: $t_{59} = 13.233$, $p < 0.001$, Q2: $t_{59} = 12.624$, $p < 0.001$). There is no significant difference between **LLM with size** and **Semantics with size** (Q1: $t_{59} = 1.841$, $p = 0.071$, Q2: $t_{59} = 0.134$, $p = 0.894$).

**Table 3: Ablation results of SceneNN dataset.**

| | affordance | place | season | size | object | KL Div. (↓) | SD (↑) | Sty. Sim. (↑) | 3D IoU (↑) | |
|---|---|---|---|---|---|---|---|---|---|---|
| | | | | | | | | | w/ scale | w/o scale |
| ours | ✓ | ✓ | ✓ | ✓ | ✓ | 0.027 | 0.386 | 0.763 | 0.858 | 0.427 |
| w/o affordance | | ✓ | ✓ | ✓ | ✓ | 0.160 | **0.453** | **0.803** | 0.886 | 0.452 |
| w/o place | ✓ | | ✓ | ✓ | ✓ | 0.110 | 0.369 | 0.681 | **0.892** | **0.453** |
| w/o season | ✓ | ✓ | | ✓ | ✓ | 0.025 | 0.386 | 0.749 | 0.864 | 0.434 |
| w/o size | ✓ | ✓ | ✓ | | ✓ | **0.015** | 0.336 | 0.695 | 0.588 | 0.204 |
| w/o object | ✓ | ✓ | ✓ | ✓ | | 0.024 | 0.387 | 0.709 | 0.859 | 0.432 |

**Table 4: Ablation results of ProcTHOR dataset.**

| | affordance | place | season | size | object | KL Div. (↓) | SD (↑) | Sty. Sim. (↑) | 3D IoU (↑) | |
|---|---|---|---|---|---|---|---|---|---|---|
| | | | | | | | | | w/ scale | w/o scale |
| ours | ✓ | ✓ | ✓ | ✓ | ✓ | 0.042 | 0.618 | 0.749 | 0.729 | 0.368 |
| w/o affordance | | ✓ | ✓ | ✓ | ✓ | 0.159 | **0.684** | 0.778 | 0.759 | 0.387 |
| w/o place | ✓ | | ✓ | ✓ | ✓ | **0.011** | 0.526 | 0.666 | **0.768** | **0.409** |
| w/o season | ✓ | ✓ | | ✓ | ✓ | 0.041 | 0.592 | 0.736 | 0.738 | 0.375 |
| w/o size | ✓ | ✓ | ✓ | | ✓ | 0.049 | 0.447 | **0.807** | 0.437 | 0.152 |
| w/o object | ✓ | ✓ | ✓ | ✓ | | 0.036 | 0.618 | 0.698 | 0.738 | 0.376 |

## 5.5 Ablation study

We conducted an ablation study to evaluate the effect of each factor. Table 3 and table 4 show the results of the ablation study. The results show that without considering affordances, although the synthesized scenes perform well in terms of scene diversity and style similarity, they do not maintain the affordances of the physical environment well. If place, season, and object are not considered, the performance of style similarity will be even worse. If the size is not considered, the 3D IoU would be relatively poor. In our evaluation, only some user input texts contain season and object information, but it still had an impact on the performance. Given that we aim to synthesize scenes that maintain the physical affordances and style that meets user demands, it is necessary to consider all factors.

## 6 DISCUSSION

### 6.1 Unlimited scenes for any physical environment.

Our method enables unlimited scene synthesis according to user demands and physical environments. In particular, if there is no user input, the method still supports the scene synthesis based on the physical environment. The sentence of user demand can be unstructured and arbitrary. It may or may not contain a place, a season, and user-specified objects. In the future, the user demand could be inferred by LLM from a simple sentence, such as *I want to rest*, according to the user's preference. Our method enables the scene synthesis for mixed reality in any physical environment as ubiquitous embodied interfaces, making it possible for future applications, such as virtual offices [23].

### 6.2 LLM-based prediction

Our method demonstrates a possibility for scene synthesis using the LLM for mixed reality that can be extended in the future. Since our scene agent predicts object properties, including affordances, location, and season, based on the LLM, the results are affected by the inference of the LLM. In the future, more accurate models can improve the performance of our method. In addition, the prompt in our method is based on text only. In the future, the multimodel prompt including images (*e.g.*, images of the virtual objects) could improve the prediction accuracy.

## 6.3 Diversity of objects

We collect a total of 350 virtual objects in the experiments. A virtual object dataset with a large scale helps synthesize scenes with more styles, to better meet user demands. It can also improve the 3D IoU and the object affordance similarity between synthesized scenes and physical environments. In addition, the virtual objects are retrieved from the dataset in our method, which can also be generated by the example-based [36, 62] or text-based generation methods [44, 64] in the future.

## 6.4 Virtual objects for physical walls

Our proposed method synthesizes scenes with virtual objects. Although our method can add virtual obstacle objects for physical walls when synthesizing scenes, we found that this is not very reasonable because the user will be surrounded by obstacles in the virtual scenes. We hope to develop a virtual scene with a broad view for users when they are situated in a limited space. In the future, the corresponding virtual objects for physical walls(*e.g.*, [59]) need more studies.

## 7 CONCLUSION

In this paper, we propose a scene agent to synthesize virtual scenes by observing the situated physical environment and demand of users, which maintains the physical affordance and user-mentioned style. The comparison results show that our method could synthesize better scenes compared with the baselines. Through the scene agent, we hope to provide users with a ubiquitous embodied interface, allowing users to access the immersive virtual environment anytime and anywhere, ensuring security while utilizing the affordance of the physical environment. This can be applied to many areas, such as virtual offices, education, and games. In the future, with the advancement of technologies such as large language models and single object generation, as well as the enrichment of virtual object datasets, our method has the potential to synthesize better scenes. Since our method can be extended based on the similarity of each factor, more factors (*e.g.*, user preferences) can be added to synthesize better scenarios.

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
