# OpenReview forum: "In Situ 3D Scene Synthesis for Ubiquitous Embodied Interfaces"
_acmmm.org/ACMMM/2024/Conference — MM2024 Poster_

### Official Review · Reviewer_L5oz · 2024-05-23

**Rating:** 5
**Confidence:** 3

**Summary:**

The paper proposes a 3D scene synthesis method driven by natural language. Language model API was applied to extract information such as physical environment and virtual objects from the input text. In addition to the text input, the synthesis method could further be constrained based on the physical affordance of the environment so that the physical object could have a corresponding virtual object in the scene. In their experiment, the proposed method was compared with other approaches comprehensively. Their ablation study also suggested that the extracted features play a role in the system.

**Strengths:**

This work is well-organized and easy to follow. The proposed system is a good demonstration of leveraging a large language model and could be useful for personalized 3D scenes. The details of the method are clearly documented. The presented figures are informative and helpful for understanding the overall design of the system. Also, they have sufficient example results to demonstrate their performance. Moreover, the comparison with other approaches indicates an improvement as well.

**Limitations:**

Some minor issues:
- Sec 4.4 “𝐼𝑜𝑈 (𝑜𝑣𝑖𝑟 |𝑜𝑝ℎ𝑦) minimizes the value of 3D Intersection over Union (IoU) between the virtual” -> should that be maximizes?
- Table 1,2 is mentioned in Sec 5.3, but some acronyms, such as SD and Sty. Sim, show up in the later section.

**Suitability:**

3

---

### Official Review · Reviewer_fE4B · 2024-05-24

**Rating:** 4
**Confidence:** 2

**Summary:**

The paper proposes a novel approach for 3D scene synthesis based on the physical environment and preferences of users. This enables users to create a virtual environment based on their real surroundings which can help to reduce accidents in VR. As the environment is based on user preferences, it can furthermore allow users to immerse themselves in their chosen environment.

**Strengths:**

The paper discusses a very interesting topic. By combining real world information with user demands, synthesized worlds can not only reduce accidents in VR, but also match user preferences and retain affordances of the real world. I think this could be an interesting solution for work in mixed reality as it can allow the user to switch to a more comfortable working environment while still protecting their safety.

The evaluation is based on quantitative and qualitative metrics. A user study confirms that the generated scenes match users wishes while comparisons between output environments and the physical environments suggest that the real-world objects are mostly covered.

The idea to combine LLMs with scene synthesis to integrate user needs seems novel and intuitive.

**Limitations:**

The evaluation is based on a RGB-D dataset and a fully synthetic dataset. It would be valuable to discuss the quality of the input depth data and the correct placement of virtual objects. How would users need to capture the environment to ensure sufficient data? Would it be possible to verify the correct coverage of synthetic scenes by users before fully entering VR to avoid accidents?

The authors should include more details on the user study.
-	Demographic information of participants (age, gender), reward for participation, how they were recruited.
-	Instructions given to participants.
-	Was this study approved by some form of ethics committee?
-	Was the study conducted in VR? Were participants able to move or were they seated? Which HMD was used? How did they interact with objects that allowed affordance?

I think the authors could clarify some points of their work. I think Fig. 2 is a bit overloaded and therefore a bit hard to understand. In addition to Fig. 2 it would be good if the authors clearly state the input and output of their method.

While the authors show many images, a short supplemental video that compares the input data and generated scenes would be very helpful.

Overall, I lean towards accepting this paper. The presented scene synthesis incorporates the novel viewpoint of combining users’ needs and physical constraints. Furthermore, the work contributes to the field of multimedia by combining depth data, virtual environments, and textual user input. I think the work could spark interesting discussions during the conference and lead to further endeavors to create individual virtual environments that suit the needs of users and match the physical surroundings.

**Suitability:**

3

---

### Official Review · Reviewer_vM3L · 2024-06-05

**Rating:** 2
**Confidence:** 3

**Summary:**

The paper proposes a framework to generate virtual scenes that blend in virtual objects at locations with obstacles in the real world. The generated virtual objects and scenes are completely based on the users prompts.

**Strengths:**

The concept of positioning virtual objects to mask physical objects and make it safer for the VR user to walk around the space is really good.
With the dataset of 350 objects, the results from the method seem to be promising for the few prompts that were provided.

**Limitations:**

The paper lacks a lot of detail about how the models were trained and their structure etc. Which makes the results not reproducible. While the supplementary material provides information that few-shot LLM was used, no further information is provided.

While the concept for the work is really good, the method itself seems very simplistic and simple probabilistic likelihood based. Leaving a lot of be desired in terms of a much more robust solution.

Since the authors are using off the shelf widely available LLM models and these models are the one decoding the users intent to provide clues on the details of the scene, it is hard to credit the results of the model for object selection and the overall scene to anything the paper is proposing.

The 3D IoU metric is used to optimize placement/scale in the adjust module described in section 4.4 and is also used in the evaluation section to highlight the superiority of placement over other methods.

One of the main aspects of the work is to cover the physical object with a virtual one and even though the 3D IoU metric is optimized for, the results are still less than desirable for the stated goal. It would be better to have a larger virtual object than the real world one for the stated motivation of the paper, but the 3D IoU metric doesn't specify it.

Considering the core premise of the exercise is to position virtual objects at locations of the real world objects so as to avoid user collisions in the real world, it would be better if the authors use a containment metric to show that the physical object would be covered by the virtual object and also use the 3D IoU to show that the virtual object is not taking way too much space.

Overall, while the idea seems novel the lack of details in many areas, use of existing LLMs and basic conditional probability for method make it feel like the first naive approach of solving the problem.

**Suitability:**

3

---

### Meta-Review · Area_Chair_Kk8V · 2024-07-01

**Recommendation:** Accept (Poster)
**Confidence:** 3

**Metareview:**

The paper presents a novel approach to generate a 3D synthetic scene based on physical environment and users movements.

Reviewers agreed that the paper is introducing an interesting topic of research in the multimedia community.  The paper is well presented.

However, many concerns have been highlighted, including the lack of important details on experiments conducted and lack of strong experimental evaluations.  The authors should report the required experimental details in the camera-ready version.  We also encourage the authors to consider the other comments and revise the paper before the camera-ready version, to strengthen the paper and its impact on the research community.